# Ethylene Oligomerization over Nickel Supported Silica-Alumina Catalysts with High Selectivity for C_{10+} Products

**Lei Chen [1,2], Guangci Li [1,*], Zhong Wang [1]** 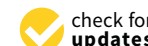 **, Shuangju Li [1,2], Mingjie Zhang [1] and Xuebing Li [1,*]**

[1]  Key Laboratory of Biofuels, Qingdao Institute of Bioenergy and Bioprocess Technology, Chinese Academy of Science, No.189 Songling Road, Laoshan District, Qingdao 266101, China; chenlei@qibebt.ac.cn (L.C.); wangzhong@qibebt.ac.cn (Z.W.); lisj@qibebt.ac.cn (S.L.); zhang_mj@qibebt.ac.cn (M.Z.)

[2]  University of Chinese Academy of Science, No.19A Yuquan Road, Beijing 100049, China

*  Correspondence: ligc@qibebt.ac.cn (G.L.); lixb@qibebt.ac.cn (X.L.); Tel./Fax: +86-532-80662757

**Abstract:** The nickel (II) loading silica-alumina under various treatments in terms of aging temperature, Si/Al ratio and activation temperature were investigated by XRD, $N_2$ adsorption-desorption, TEM, UV-Vis, $NH_3$-TPD and XRF and then applied to catalyze the ethylene oligomerization. High aging temperature, low Si/Al ratio and high activation temperature were beneficial to high selectivity for $C_{10+}$ products because of a reasonable match between Ni active sites and acid sites, high Ni loading content and less octahedral coordination $Ni^{2+}$ species, respectively. Ni loading content was more important than the number of acid sites for high yield of $C_{10+}$ products, and less octahedral coordination $Ni^{2+}$ species favored less by-products produced at high reaction temperature. In addition, other experimental conditions, such as reaction temperature, weight hourly space velocity (WHSV) and nickel precursor were discussed in the paper.

**Keywords:** ethylene oligomerization; $Ni-SiO_2-Al_2O_3$; Si/Al ratio; synthesis aging temperature; octahedral coordination $Ni^{2+}$

## 1. Introduction

The oligomerization of light olefins is an important route for the production of linear and branched higher olefins, which can be used in the manufacture of detergents, petrochemicals, oil additives, high-octane ecological gasoline, etc. [1,2]. Ethylene, with huge production worldwide, is the raw material for a wide range of chemical products and intermediates. Industrial reactions of ethylene include in order of scale polymerization, oxidation, halogenation, alkylation, hydration, oligomerization and hydroformylation [3–6]. Ethylene oligomerization is of considerable academic and industrial interest because it is one of the major processes for production of linear and branched higher olefins, which are components of plastics ($C_4$–$C_6$ in copolymerization), plasticizers ($C_6$–$C_{10}$ through hydroformylation), lubricants ($C_{10}$–$C_{12}$ through oligomerization) and surfactants ($C_{12}$–$C_{16}$ through arylation/sulphonation) or starting materials for other important chemicals, such as propylene, alcohols, amines and acids [1,7–13]. Among those higher olefins products, $C_{10+}$ is very desirable for jet fuel application [14–16].

The $C_{10+}$ olefins produced from ethylene can be fulfilled by using homogeneous [17–20] and heterogeneous catalysts [21–23] where heterogeneous catalysts have been extensively explored because of easily separation from the product and better reusability. Among those heterogeneous catalyst systems, Ni-base catalysts have attracted much attention because of high activity and selectivity towards the $C_{10+}$ olefins [1,24–30]. The oligomerization on Ni-based catalysts coupled with co-oligomerization reactions involving the primary olefins over an acid catalyst is favorable for obtaining the olefins with

$C_{10+}$ chain. In order to illustrate the role of nickel sites and acid sites in this reaction, the total reaction pathways were proposed, as shown in Scheme 1 [6].

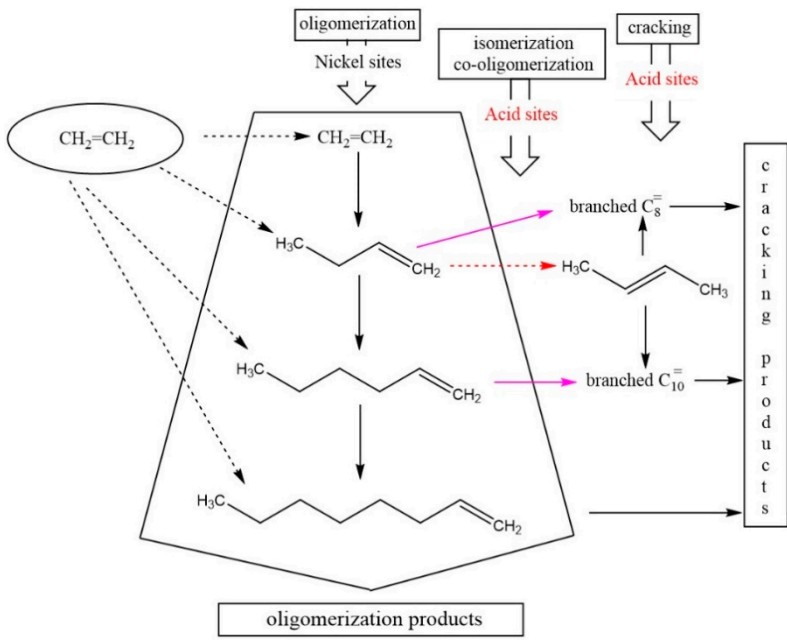

**Scheme 1.** Reaction pathway in the ethylene oligomerization process[6].

The first reaction is based on the coordination chemistry on nickel sites. They act as active sites for both the initial oligomerization of ethylene and further oligomerization reactions involving butene–ethylene coupling, leading to linear olefins of medium-chain length. The second one is based on the acid catalysis. Over acid sites, the $C_4$ and $C_6$ olefins can be consumed through co-oligomerization reactions (mechanism involving carbenium ions), leading to the formation of octenes or higher branched olefins, respectively. These reactions are essentially favored by a stronger acidity or/and a higher acid sites concentration and higher reaction temperatures. The same factors are responsible for the isomerization of the initial product (1-butene, 1-hexene, etc.). The $C_4$–$C_{10}$ oligomers can be involved in further acid catalyzed reactions, leading to the formation of heavy hydrocarbons which are responsible for pore blocking and catalyst deactivation. The third type of reaction, occurring under severe conditions and involving the acid sites, consists in the cracking of the primary and secondary oligomers and H transfer.

In most previous studies, the active Ni species are often loaded on the different supports, such as silica[17], silica–alumina [6,24] and zeolites [31–34], to improve the dispersion of Ni species. Meanwhile, the activity and selectivity towards the $C_{10+}$ olefins are also affected with the change in the acid sites and the porosity of the supports. For example, Ng et al. [33] investigated the effect of NaY zeolites supports by the acid and base treatment and the calcination temperature, and found that Ni species and acid sites were necessary for the oligomerization of ethylene. Lallem and et al. [28] reported that the Ni-exchanged MCM-36 zeolite exhibited higher activity and stability than the Ni-exchanged MCM-22 zeolite because the mesoporous nature of the MCM-36 zeolite facilitates the diffusion of larger oligomers formed during the reaction.

Apart from the zeolites and mesoporous silica, silica–alumina support was the earlier candidate because of their merits of easy synthesis and cheapness. Compared with zeolite materials, silica–alumina support has the moderate strength acid sites that benefit to lower the extent of over-polymerization of ethylene, thus inhibiting the coke formation. Moreover, silica–alumina support also has the interparticle mesopores which facilitate the larger oligomers diffusion. Heveling et al. [34] employed this catalytic system to obtain products in the $C_4$–$C_{20}$ range from ethylene and found these types of catalysts were extremely stable, showing no detectable drop in conversion after 108 days. Previous

studies on Ni-loading silica–alumina mainly investigated the effect of reaction conditions [34] (reaction temperature and gas pressure) and the reactor type (fixed-bed reactor [34] and the slurry reactor [24]). In this work, we focused on the effect of the modification of the silica–alumina support in terms of aging temperature, the ratio of Si/Al and activation temperature and the $C_{10+}$ yield was employed to evaluate these effects. In addition, we investigated the effects of other experimental conditions on the experimental results, such as reaction temperature, weight hourly space velocity (WHSV) and nickel precursor.

## 2. Results and Discussion

### 2.1. The Physicochemical Properties of Ni/Si-Al Catalysts

Figure 1 shows the XRD patterns of as-obtained Si-Al support and Ni/Si-Al catalyst with different Si:Al ratios treated at different aging temperatures. For all samples, including Si-Al support and Ni/Si-Al catalysts, there was only a broad peak locating at around 22.5° and no sharp peaks attributed to $Al_2O_3$ were observed, indicating its amorphous structure and that they were composed of small particles with poor crystallinity. Accordingly, the ratio of Si/Al and aging temperature cannot influence the crystalline structure of Si-Al support and Ni/Si-Al catalyst under present synthesis condition.

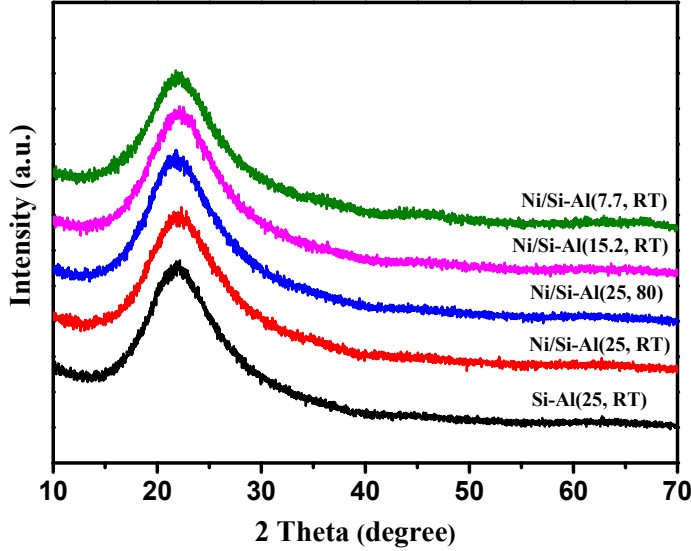

**Figure 1.** The XRD patterns of prepared catalysts at different conditions.

After loading Ni onto Si-Al support, there was no obvious peaks attributed to NiO in XRD patterns for all Ni/Si-Al catalysts, but XRF texts (as shown in Table 1) verified that Ni species does exist on the support surface. The diffraction peaks for Ni species were not found in XRD patterns, resulted from the lower amount of Ni loading by ion-exchange method and the well dispersity of Ni on the support surface, which is beneficial to the catalytic performance.

**Table 1.** The catalysts prepared by $Ni(NO_3)_2$-sodium silicate-$NaAlO_2$ and their physical properties.

| Catalysts | Ni (%) [a] | Surface Area ($m^2/g$) [b] | $V_{meso}$ ($m^3/g$) [b] | Acid Amount (mmol/g) [c] |
|---|---|---|---|---|
| Ni/Si-Al (25, RT) | 1.36 | 180 | 0.71 | 0.5 |
| Ni/Si-Al (25, 80) | 1.34 | 219 | 0.79 | 0.49 |
| Ni/Si-Al (15.2, RT) | 1.45 | 107 | 0.41 | 0.41 |
| Ni/Si-Al (7.7, RT) | 1.94 | 97 | 0.40 | 0.37 |

[a] determined by XRF. [b] determined by $N_2$ adsorption-desorption test (Table S1). [c] determined by $NH_3$-TPD (Figure S1 and Table S2).

Figure 2 shows the morphology of the Ni/Si-Al catalysts with different Si/Al ratio or aging temperature. The resultant sample shows that Ni/Si-Al catalysts at different aging temperatures had a uniform spherical morphology with the particle size in the range of 20–30 nm, while with increasing the aging temperature, there was no obvious change in the morphology and particle size (Figure 2A and B, Ni/Si-Al (25, RT) vs. Ni/Si-Al (25, 80)). However, when the ratio of Si/Al was decreased from 25 to 12.5, the single spherical particles disappeared and adjacent particles began to grow and combine into worm-like particles with the size of 50 nm and more (Figure 2A,C, Ni/Si-Al (25, RT) vs. Ni/Si-Al (12.5, RT)). As the ratio of Si/Al was further decreased, the worm-like particles grew into larger aggregates (Figure 2D, Ni/Si-Al (7.7, RT)).

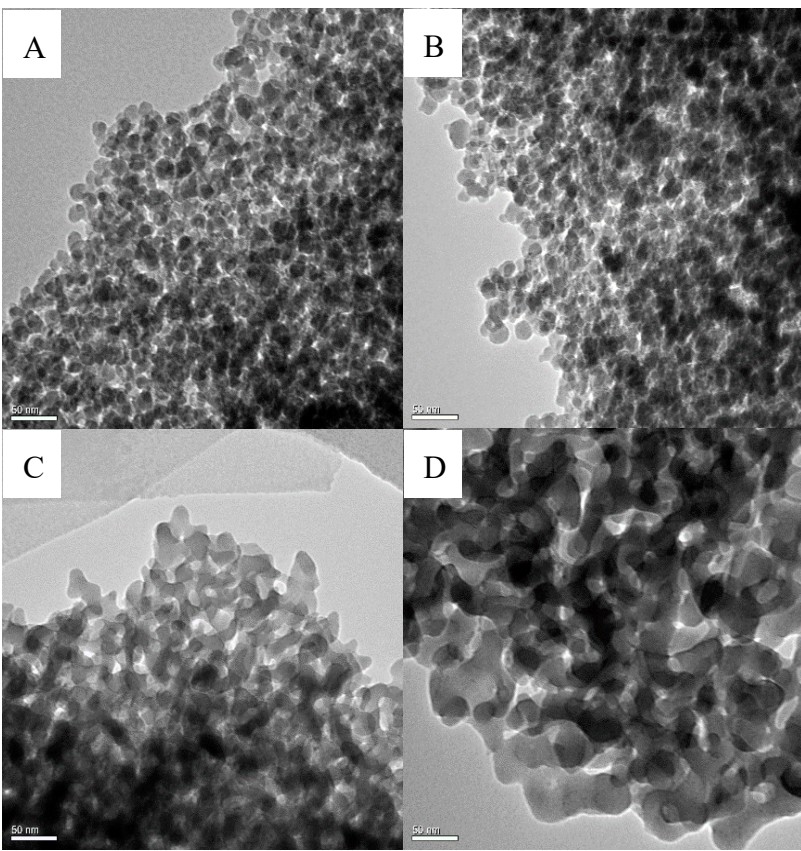

**Figure 2.** TEM images of prepared catalysts: (**A**) Ni/Si-Al (25, RT); (**B**) Ni/Si-Al (25, 80); (**C**) Ni/Si-Al (15.2, RT); (**D**) Ni/Si-Al (7.7, RT).

The change in morphology would result in some differences in textural properties of Ni/Si-Al catalysts. The increase in particle size inevitably resulted in the decrease of the surface area, as shown in Figure 3 and Table 1. Nitrogen sorption isotherms for the Ni/Si-Al (x, y) are shown in Figure 3. According to the International Union of Pure and Applied Chemistry (IUPAC) classification, all isotherms are of type IV [35]. Their sharp capillary condensation steps are indicative of a narrow mesopore size distribution. The highly parallel adsorption and desorption branches in the hysteresis loop indicate that the sample possesses uniform mesopores. Despite the similar shape of isotherms, the adsorption amount of the three samples is different, which means some differences in their textural properties. Based on the results calculated by Brunauer-Emmett-Teller (BET) and Barrett-Joyner-Halenda (BJH) methods, as the ratio of Si/Al decreased, the specific surface area decreased in order ($180 > 107 > 97$ m$^2$/g) as well as the pore volume ($0.71 > 0.41 > 0.40$ cm$^3$/g) (Table 1). These changes would affect the dispersion of Ni and corresponding catalytic activity. Compared with Ni/Si-Al (25, RT), Ni/Si-Al (25, 80) catalyst had higher surface area and Vmeso, indicating that treating the catalysts at higher temperature was benefited to an increase in the surface area and Vmeso.

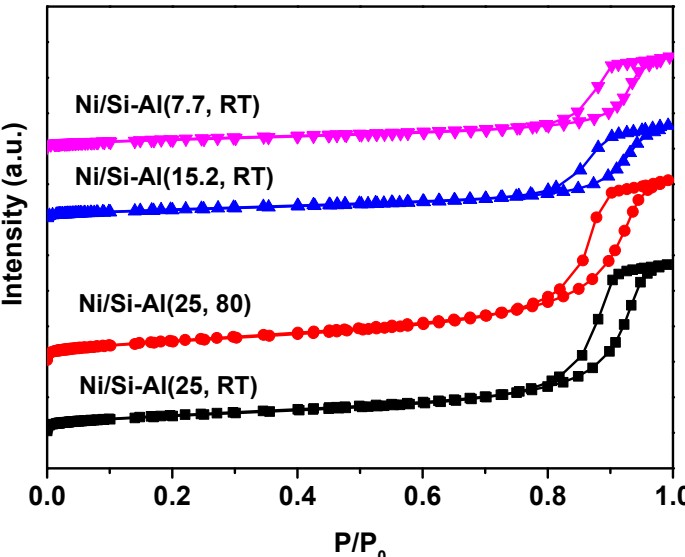

**Figure 3.** N$_2$ adsorption-desorption isotherms at 77 K determined on Ni/Si-Al(x,y).

For the acidity property, the total acid amount remained constant after treating at different temperature (0.5 mmol/g in Ni/Si-Al (25, RT) vs. 0.49 mmol/g in Ni/Si-Al (25, 80)), but decreased with the Si:Al ratio (Table 1, Ni/Si-Al (25, RT) > Ni/Si-Al (15.2, RT) > Ni/Si-Al (7.7, RT)). It has been well documented that the acid sites on the surface of silica–alumina were mostly from the alumina addition [6,36–38]. This unexpected decreased acid amount was presumably attributed to the increase of Ni loading content, which may occupy the acid sites (Table 1). In each sample, there was no obvious change in the acid strength, indicating that the Si/Al ratio had no effect on the distribution of acid strength (Table S1). Furthermore, since Na$^+$ ions are exchanged with Ni$^{2+}$ ions, the loading content of Ni species is proportional to the content of Na$^+$ ions (Table 1, Ni/Si-Al (7.7, RT) > Ni/Si-Al (15.2, RT) > Ni/Si-Al (25, RT)).

*2.2. Ethylene Oligomerization Reaction Tests*

As mentioned above, the catalyst property and reaction condition will affect the properties of ethylene oligomers remarkably. Thus, the effects of several factors on ethylene oligomerization were investigated respectively, including reaction temperature, WHSV, Ni precursor, aging temperature, the ratio of Si/Al and pretreatment temperature of catalyst.

2.2.1. The Effect of Reaction Temperature

In order to investigate the effect of reaction temperature on the ethylene oligomerization, a series of experiments at different temperatures were carried out. The gas phase product at all reaction temperatures, the content of butene is above 98%, considering the composition of the liquid phase product together, it can be concluded that the conversion of ethylene is above 98%. As shown in Figure 4, it can be seen that the olefins with low carbon number (C$_6$ and C$_8$) are predominant in the liquid product when the reaction proceeded below 200 °C.

2.2.2. The Effect of Weight Hourly Space Velocity (WHSV)

Space velocity is one of important factors for heterogeneous catalysis reaction, which represents the time of reactant contact with catalyst. For investigating the effect, the reaction was conducted with WHSV in the range from 2.0 to 3.72 h$^{-1}$, and the conversion of ethylene is above 95% with different yield of liquid, the result is shown in Figure 5. It was found that the liquid yield is dependent on WHSV. When WHSV was controlled to be 2.0 h$^{-1}$, the liquid yield was about 40%. Although reducing WHSV could prolong the time of ethylene contact with catalyst and improve oligomerization reaction,

long-time contact would lead to excessive oligomerization of olefins into carbon, inducing low liquid yield. When WHSV was increased more than 3.0 h$^{-1}$, the liquid yield further decreased. This may be due to the time of contact of ethylene with catalyst being too short, which results in that the as-obtained short-chain olefins with detached from catalyst before further reaction. When WHSV was 2.8 h$^{-1}$, the highest liquid yield was obtained.

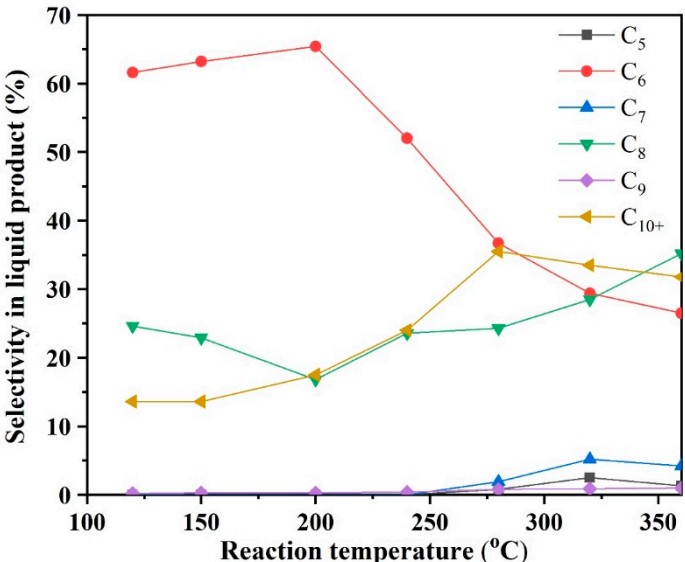

**Figure 4.** Effect of reaction temperature on the selectivity in the liquid product. Reaction conditions: Si/Al ratio: 25; pressure: 3.5 MPa; WHSV: 2.0 h$^{-1}$; Ni precursor: Ni(NO$_3$)$_2$; aging temperature: room temperature; pretreatment temperature: 300 °C. As the reaction temperature was increased, the contents of C$_6$ and C$_8$ decreased gradually, accordingly, the contents of C$_{10+}$ increased obviously. When the temperature was beyond 300 °C, the olefins with odd carbon number (C$_5$, C$_7$ and C$_9$) appeared, indicating that cracking reaction took place. Thus, high temperature is favorable for the formation of C$_{10+}$ olefins, but too high temperature will affect the liquid yield because of cracking reaction.

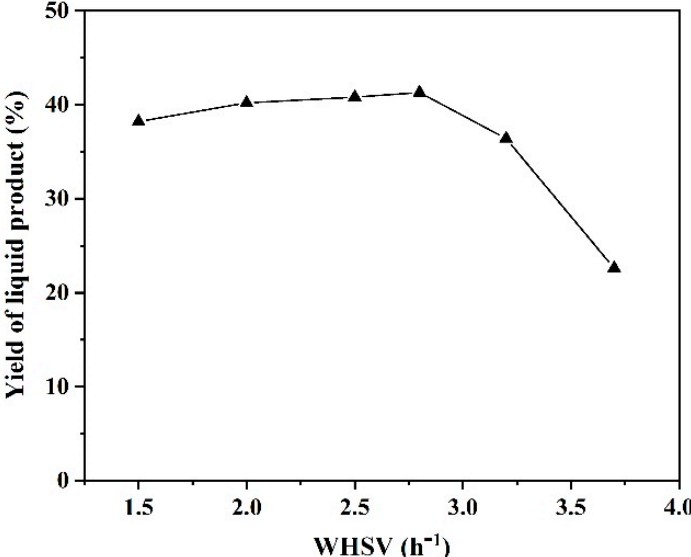

**Figure 5.** Effect of WHSV on the yield of liquid product. Reaction conditions: reaction temperature: 120 °C; Si/Al ratio: 25; pressure: 3.5 MPa; Ni precursor: Ni(NO$_3$)$_2$; aging temperature: room temperature; pretreatment temperature: 300 °C.

### 2.2.3. The Effect of the Ni Precursors

To understand how the Ni precursor affects the Ni/Si-Al activity, two kinds of catalysts were prepared by using $Ni(NO_3)_2$ and $NiCl_2$ as the Ni precursor, respectively. As seen from Figure 6, the activity of Ni/Si-Al derived from $Ni(NO_3)_2$ is better than that of Ni/Si-Al derived from $NiCl_2$ at low reaction temperature. When the temperature exceeded 280 °C, Ni/Si-Al derived from $NiCl_2$ exhibits higher activity. This result is relative to the Ni precursor, especially anions. The detailed effect mechanism needs further investigation.

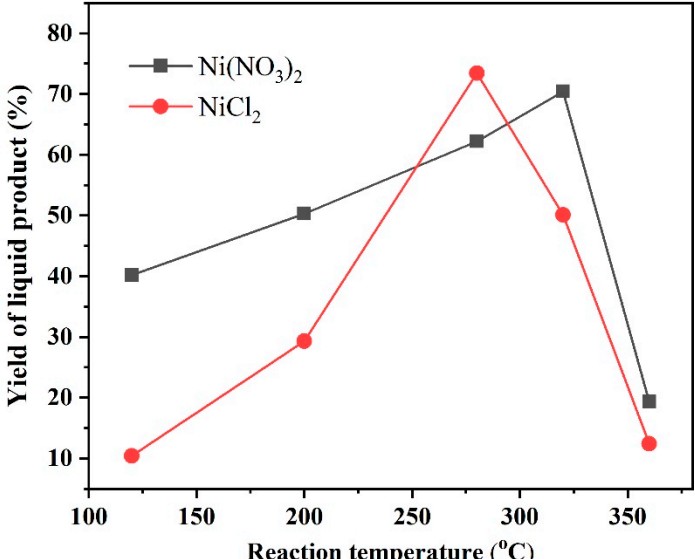

**Figure 6.** Effect of Ni precursor on the yield of liquid product. Reaction conditions: Si/Al ratio: 25; pressure: 3.5 MPa; WHSV: 2.0 $h^{-1}$; aging temperature: room temperature; pretreatment temperature: 300 °C.

### 2.2.4. The Effect of Aging Temperature

Figure 7 showed the result of as-obtained catalysts catalyzed ethylene oligomerization, where the yield of $C_{10+}$ products was used as an indicator for evaluating their performance. With increasing the reaction temperature, the yield of $C_{10+}$ products obtained from both Ni/Si-Al (25, RT) and Ni/Si-Al (25, 80) first increased (below 280 °C) and then decreased (above 280 °C). This was because increasing temperature favored accelerating the rate of ethylene oligomerization, but too high a temperature led to undesirable side reactions, resulting in low selectivity for $C_{10+}$ products. Compared with Ni/Si-Al (25, RT), Ni/Si-Al (25, 80) catalyst could deliver higher $C_{10+}$ products at all reaction temperatures used, indicating that aging the catalysts at high temperature was beneficial to its catalytic performance.

For catalysts with little difference in acid property and similar Ni loading content (As shown in Table 1), the reaction results were different. The reason is that as the aging temperature increases, the degree of condensation of the Si-Al support increases, that is, the dispersion of Si-O-Al is more uniform. Considering the mutual combination of Na and Al in the support, the dispersibility and the spatial order of Na is improved, and the loading of Ni is obtained by the exchange of Ni and Na, so the spatial order of Ni is improved [6,23,25]. One result of uniform dispersion of nickel is a reasonable match between Ni active sites and acid sites. The combination of a reasonable match between Ni active and acid sites can be seen as a "more efficient active site" for ethylene oligomerization, consistent with previous reports [28]. Figure 8 intuitively showed the benefit of high dispersity of Ni sites. In an oligomerization catalytic cycle, there will be (I) desorption of primary oligomers and (II) re-adsorption at acid sites for secondary oligomerization events, "more efficient active site" include Ni active sites and acid sites on the catalysts surface, can facilitate the transfer of reaction intermediates between two

active sites, so that the intermediate products continue to react to give longer carbon chain products. That is, the percentage of the $C_{10+}$ products increased the oligomerization products.

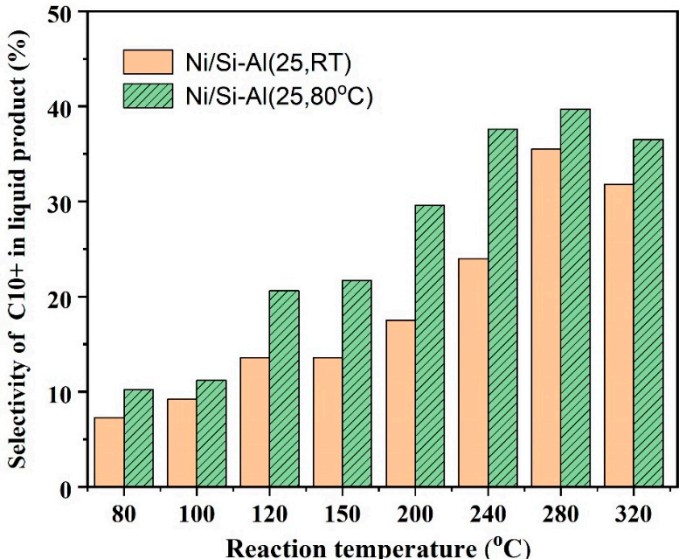

**Figure 7.** $C_{10+}$ product of ethylene oligomerization catalyzed by Ni/Si-Al (25, RT) and Ni/Si-Al (25, 80) at different reaction temperatures. Reaction conditions: pressure: 3.5 MPa; WHSV: 2.0 h$^{-1}$; Ni precursor: Ni(NO$_3$)$_2$; aging temperature: room temperature; pretreatment temperature: 300 °C.

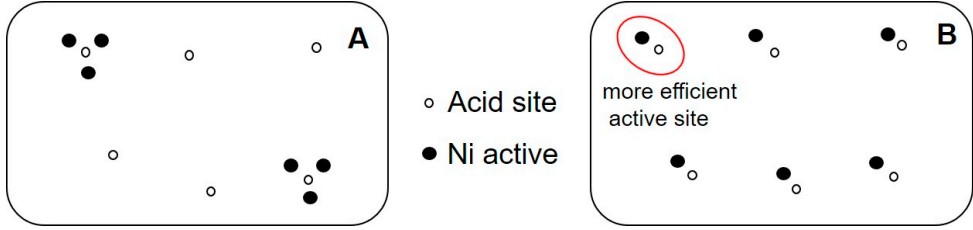

**Figure 8.** Match scheme of nickel and acid sites of (**A**) low nickel dispersed degree and (**B**) high nickel dispersed degree.

2.2.5. The Effect of the Ratio of Si/Al

As discussed above, changing the ratio of Si/Al of these catalysts could affect its properties involving morphology, structure and textural properties. It is believed that these properties are related to the catalytic performance of the Ni/Si-Al catalyst.

The catalytic performances of these catalysts with different Si/Al ratios are shown in Figure 9. At the same reaction temperature, decreased Si/Al ratio is favorable for the formation of $C_{10+}$ olefins when the reaction temperature is below 240 °C. With increasing the reaction temperature, the yield of $C_{10+}$ products from Ni/Si-Al (25, RT) and Ni/Si-Al (15.2, RT) continually increased, while for Ni/Si-Al (7.7, RT) catalyst it first increased and reached the maximum value at 240 °C and then decreased slowly when the reaction temperature was below 360 °C. There was a sharp decrease in the proportion of $C_{10+}$ in liquid product at 360 °C, according to the above discussion, we can speculate that the reaction temperature is too high and cracking reaction occurred.

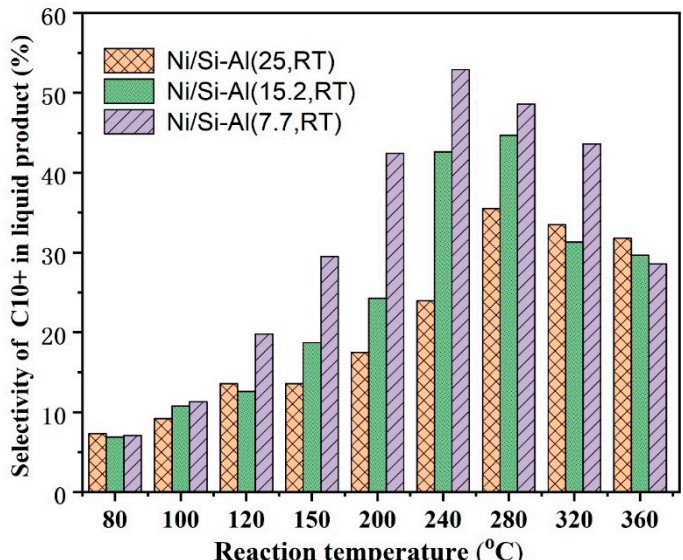

**Figure 9.** $C_{10+}$ product of ethylene oligomerization catalyzed by Ni/Si-Al (25, RT), Ni/Si-Al (15.2, RT) and Ni/Si-Al (7.7, RT). Reaction conditions: Pressure: 3.5 MPa; WHSV: 2.0 $h^{-1}$; Ni precursor: $Ni(NO_3)_2$; pretreatment temperature: 300 °C.

Moreover, the yield of $C_{10+}$ products from Ni/Si-Al (7.7, RT) was much higher than that from Ni/Si-Al(25, RT) and Ni/Si-Al(15.2, RT) at the temperature range from 80 to 240 °C. These results indicated that the lower Si/Al ratio was in favor of obtaining high selectivity for $C_{10+}$ products at a lower reaction temperature. Combining the characterization results of the catalysts, it can be found that the physical properties of samples exhibit a good relationship with the content of the $C_{10+}$ olefins. We conclude that the specific surface area directly influences the dispersion of Ni species locating on the silica–alumina surface. As described earlier, because of the decrease in the Si/Al ratio, the number of $Na^+$ ions increase during the preparation of silica–alumina. Since $Na^+$ ions are exchanged with $Ni^{2+}$ ions, the loading content of Ni species is proportional to the content of $Na^+$ ions. As a result, more Ni species must disperse on the silica–alumina with lower surface area, which means the density of Ni species on the surface of support increased as the Si/Al ratio decreased. Thus, it was thought high-density surface nickel could promote the conversion of ethylene into $C_{10+}$ olefins.

When compared with Ni/Si-Al (25, RT) and Ni/Si-Al (15.2, RT), Ni/Si-Al (7.7, RT) had lower acid amounts but higher Ni loading content (Table 1), indicating higher Ni loading content played a more critical role than the acid amount in obtaining higher selectivity for $C_{10+}$ products and lowering the reaction temperature.

### 2.2.6. The Effect of Activation Temperature

The catalyst Ni/Si-Al (7.7, RT) was chosen to investigate the effect of activation temperature because of its high catalytic performance based on the above results. Figure 10 showed the UV-vis spectrum of Ni/Si-Al (7.7, RT) samples activated at different temperatures (no activation, 200 °C, 400 °C). The peak at 270 nm was attributed to the three-coordinate $Ni^{2+}$ ($Ni_{3c}^{2+}$), which was also observed by Klier [39] and Cornet [40]. This 270 nm peak was also related to the charge between $O^{2-}$(2p) and $Ni^{2+}$(3d), which was different from the agglomerated NiO [41], indicating the well-dispersion of NiO on the support without agglomeration. This was in accordance with the XRD results where no NiO peaks were observed. There was no peak shift in 270 nm, indicating that the status of $Ni_{3c}^{2+}$ had nothing to do with the activation temperature.

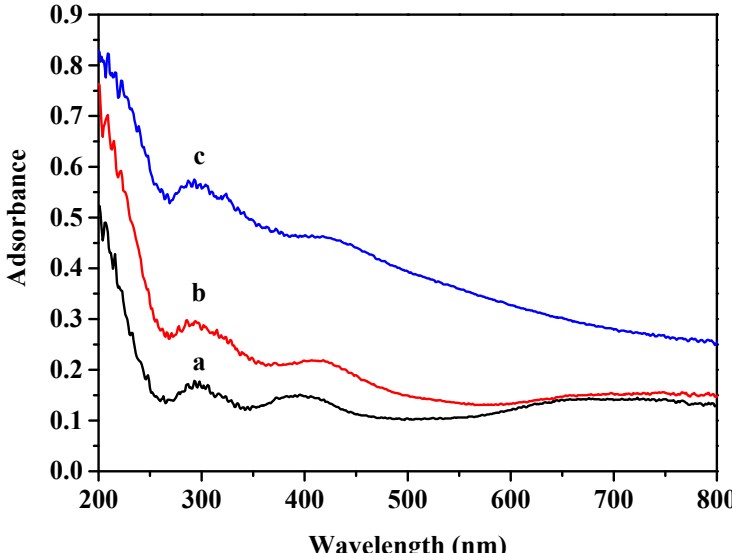

**Figure 10.** UV-Vis spectrum of Ni/Si-Al (7.7, RT) activated at (**a**) room temperature, (**b**) 200 °C and (**c**) 400 °C, respectively.

However, the shoulder peak at 414 nm attributed to octahedral coordination $Ni^{2+}$ [24,42–44] moved to a long wavelength gradually with an increased activation temperature. The intensity of this peak also decreased with increasing activation temperature. These results implied that the high activation temperature could affect the octahedral coordination $Ni^{2+}$ in the way of reducing its amount, which was consistent with previous reports where the increased activation temperature could make octahedral coordination $Ni^{2+}$ diffuse into support to form tetrahedral coordination $Ni^{2+}$ [25,45–47].

Figure 11 showed the catalytic performance of Ni/Si-Al (7.7, RT) catalysts activated at room temperature, 200 and 400 °C. With the reaction time, the $C_{10+}$ yield profile showed a similar trend where it first increased and then decreased. Unexpectedly, when the catalysts activated at room temperature, a sharp decrease (form 24.4% to about 4%) in the $C_{10+}$ yield profile at high temperature (above 240 °C). Compared with the sample that activated at room temperature, the sample activated at 200 and 400 °C delivered more $C_{10+}$ product at all the temperature range used except 80 and 100 °C. These results implied that the activation process done at relatively high temperature improves the selectivity for $C_{10+}$ products. When compared with the catalyst activated at 200 °C, higher $C_{10+}$ products were obtained in the catalyst activated at 400 °C, the same conclusion emerges when compared with the previous data (as shown in Figure 9). This superior property in $C_{10+}$ selectivity of the catalyst activated at 400 °C mainly resulted from the decreased octahedral coordination $Ni^{2+}$ (Figure 10). At a high reaction temperature above 240 °C, the catalyst activated at 400 °C was also superior to that activated at 200 °C in obtaining $C_{10+}$ products. High $C_{10+}$ products at high temperature indicated that the undesirable less side reactions occurred. Such a finding implied that more tetrahedral coordination $Ni^{2+}$ was helpful to suppress the side reactions when reaction proceeded at high temperature because the catalyst activated at 400 °C had more content of tetrahedral coordination $Ni^{2+}$ than that activated at 200 °C.

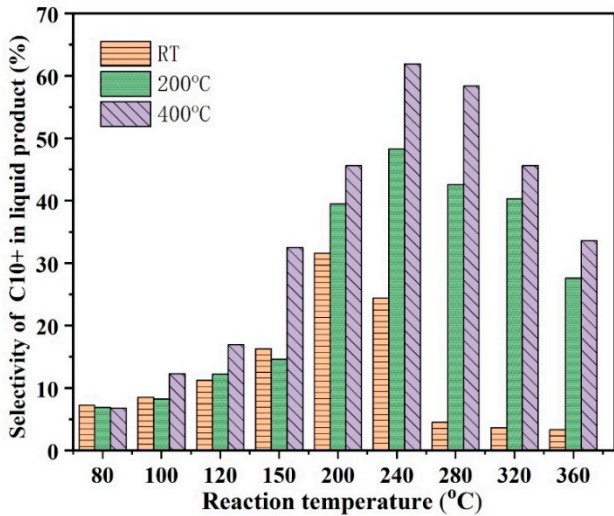

**Figure 11.** $C_{10+}$ product of ethylene oligomerization catalyzed by Ni/Si-Al (7.7, RT) activated at 200 and 400 °C, respectively. Reaction conditions: pressure: 3.5 MPa; WHSV: 2.0 h$^{-1}$; Ni precursor: Ni(NO$_3$)$_2$; aging temperature: room temperature.

## 3. Materials and Methods

### 3.1. Preparation of Silica–Alumina Supports

The type of silica–alumina support was prepared by the sol-gel method according to a previous report [34] with certain modification. Typically, a certain amount of NaAlO$_2$ was firstly dissolved into 0.26 M NaOH solution as the solution A. Then, 39.8 mL of sodium silicate solution was diluted by 217 mL of water as the solution B. After that, 45.6 mL of solution A was added into the solution B, then 86 mL of 1.4 M nitric acid was added into above mixture with stirring for 0.5 h. Finally, pH was adjusted to 9 using NaOH solution. The mixture was further aged at room temperature or 80 °C without stirring for 3 days, and then precipitation was filtered, washed with deionized water until the wash water was neutral and dried at 80 °C overnight and calcined at 550 °C for 3 h. The Na$^+$-form of the solid silica–alumina thus was synthesized.

### 3.2. Preparation of Nickel Supported Silica–Alumina by Ion-Exchange Method

Ni loading catalyst was prepared by ion-exchanging the obtained silica–alumina powder with an aqueous solution of nickel metal salts at 60 °C with stirring for 6 h. The molar concentration of the nickel solution was 8 mol/L and using three moles of nickel (II) for every two moles of aluminum in the silica–alumina support. After that, the green solid was filtered, washed with deionized water for several times and dried at 110 °C for 6 h, and then nickel supported silica–alumina catalysts were obtained. The synthesized catalysts were labeled as Ni/Si-Al(x, y), where x represents the Si/Al ratio and y is the aging temperature in which RT means room temperature.

### 3.3. Characterizations

X-ray diffraction (XRD) measurements were carried out on a Bruker AXS-D8 Advance powder diffractometer using Cu Kα radiation (40 kV, 40 mA), with a step size of 0.02° (2θ) and 2 s per step over the 2θ range from 10° to 70°. The nitrogen adsorption–desorption isotherms at 77 K were determined on an Autosorb-6B instrument, using nitrogen of 99.999% purity. Transmission electron microscopy (TEM) studies were carried out on a Hitachi H-7650 electron microscope with an accelerating voltage of 100 kV. Temperature-programmed desorption of ammonia (NH$_3$-TPD) for acidity analysis was carried out in a Micromeritics 2920TR Chemisorption Analyzer. One hundred and fifty mg of sample was pretreated at 873 K in Ar flow for 2 h. After cooled down to room temperature, a pure NH$_3$ was adsorbed for 2 h. Desorption of NH$_3$ was monitored in the temperature range of 373–773 K with the heating

rate of 10 K/min. Uv-vis absorption characterization was tested using Hitachi u-4100, the scanning wavelength range is 200–900 nm and the scanning rate is 300 nm/min. XRF was used to determine the amount of element in the catalysts using Panalytical Axios PW 4400-X-ray fluorescence spectroscopy.

*3.4. Catalytic Experiment*

The catalytic experiment was performed in a fixed-bed micro-reactor. Prior to each experiment, the catalysts were firstly activated at 300 °C and 50 mL/min He before introducing ethylene. The reactor was then heated to a certain temperature with introducing ethylene (3.5 MPa and $2h^{-1}$ of WHSV). Prior to the reaction, the ethylene was absorbed by the mixture of NaOH, 4A zeolite and anhydrous $CaCl_2$ for purification. The reaction system is shown in Figure S1.

The gas products were analyzed with on-line gas chromatographs (Agilent 7890A) equipped with HP-PLOT/Q capillary column (30 m × 0.32 mm × 20 μm) and FID (flamed ionized detector), while liquid products were analyzed with HP-INNOWAX capillary column (30 m × 0.25 mm × 0.25 μm). The compositions of products were identified using Agilent 5975 inert XL MSD GC/MS instrument. The conversion of ethylene and its selectivity to olefins were calculated by the area of FID signal of each compound.

## 4. Conclusion

Three kinds of amorphous silica–alumina with different Si/Al ratios and the corresponding Ni/Si-Al catalysts were prepared by the ion-exchange method. Under all the experimental conditions employed in this study, the conversions of ethylene were over 95%. The aging temperature, Si/Al ratio and activation temperature affected the catalytic performance of Ni loading silica–alumina catalysts through making changes in the acid sites, Ni loading amount, surface area and the type of Ni species rather than the amorphous structure. High Ni loading content was more important for higher $C_{10+}$ production than the surface area and acid mount while increased tetrahedral coordination $Ni^{2+}$ tend to suppress the side reactions when catalysts were used at high temperature. The better catalytic performance for obtaining more $C_{10+}$ products can be achieved in catalysts with low Si/Al ratio, high aging and activation temperature. In addition, the effects of several reaction conditions on the ethylene oligomerization activity of Ni/Si-Al were investigated. It is found that increasing reaction temperature, aging temperature, and pretreatment temperature are favorable for the formation of $C_{10+}$ olefins.

**Supplementary Materials:** Supplementary materials are available online at http://www.mdpi.com/2073-4344/10/2/180/s1. Table S1: Textural properties of Ni/Si-Al(x, y) determined by $N_2$ adsorption-desorption, Table S2. The acid distribution of Ni/Si-Al(x, y) from $NH_3$-TPD, Figure S1. $NH_3$-TPD profiles of Ni/Si-Al(x, y), Figure S2. Schematic diagram of ethylene oligomerization unit.

**Author Contributions:** Conceptualization, L.C., G.L. and X.L.; Data curation, L.C.; Formal analysis, L.C.; Funding acquisition, X.L.; Investigation, L.C.; Resources, G.L., Z.W., M.Z. and S.L.; Software, S.L.; Supervision, X.L.; Visualization, L.C.; Writing–original draft, L.C.; Writing–review & editing, L.C. All authors have read and agree to the published version of the manuscript.

**Funding:** Financial supports from the National Natural Science Foundation of China (No. 21676287, 21761132006) are gratefully acknowledged.

**Conflicts of Interest:** The authors declare no conflict of interest.

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
