# Peer review of "Ethylene Oligomerization over Nickel Supported Silica-Alumina Catalysts with High Selectivity for C10+ Products"

_catalysts, doi:10.3390/catal10020180_

Round 1
Reviewer 1 Report
This review concerns the article entitled: Ethylene oligomerization over nickel supported silica-alumina catalysts with high selectivity for C10+ olefin products (Manuscript ID: catalysts-706186).
In the submitted work, the ethylene oligomerization was performed using heterogeneous nickel catalysts, obtained by ion exchange method, on three types of silicon-aluminum supports, which differ in Si/Al mole ratio as well as aging and activation temperatures and various nickel content. Authors elaborated optimal oligomerization conditions, which result in 60% yield of C10+ olefins at 95% ethylene conversion.
In my opinion the manuscript should be accepted after minor revision. The detailed remarks are given below:
In subsection 3.1 (lines 159-163), the results described and presented in Table 1 (amongst other surface area and mesopores volume) are strictly connected with the data obtained from isotherm N1 adsorption-desorption and the results calculated by BET nad BJH methods, which are presented at the end of this section (lines 178-186). In my opinion these fragments should be joined together. This enables better explanation of various nickel loading on the supports. Furthermore, I suggest to place here an explanation concerning some differences in the Ni content as well as acid sites on the supports. Presently, these explanations are placed both in subsection 3.2.4 and 3.2.5. Moreover, there is lack of more detailed reason why the figure 8 was placed in the text as well as references to the schemes A and B on this figure 8. Please, explain clearly whether table 1 concerns the data for the catalysts obtained by incorporation on silica-alumina powder Ni(NO3)2 or NiCl2. Please describes whether the type of applied nickel sites influences the texture and physical and chemical properties of the catalyst or not. Please explain, whether the effect of reaction temperature on the selectivity of product and WHSV on yield of product was investigated on the catalyst Ni/Si-Al (25, RT) or Ni/Si-Al (25, 80). Why this catalyst was chosen, although the best catalyst for obtaining C10+ olefins was that at Ni/Si-Al (7.7 RT)? The references should be written in the same style.Author Response
Dear Reviewer :
Thank you for your comments concerning our manuscript entitled “Ethylene oligomerization over nickel supported silica–alumina catalysts with high selectivity for C10+ products” (ID: catalysts-706186).
We gratefully appreciate the valuable comments and suggestions from the editor and reviewers, which significantly helped improve the quality of our work. We carefully addressed all the comments point-by-point and significantly revised our manuscript (using "Track Changes" function or marked in blue) and the SI accordingly.
Please see the attachment.
Thank you and best regards.
Yours sincerely,
Guangci Li
Corresponding author:
Email: ligc@qibebt.ac.cn

Reviewer 2 Report
The work of Chenet. al. is a solid piece of work thta deserves publication in Catalysts as is.
Author Response
Dear Reviewer:
Thank you for taking the time to review my manuscript.“Ethylene oligomerization over nickel supported silica–alumina catalysts with high selectivity for C10+ products” (ID: catalysts-706186).Manuscript ID: catalysts-706186
Thank you and best regards.
Yours sincerely,
Guangci Li
Corresponding author:
E mail: ligc@qibebt.ac.cn
Reviewer 3 Report
Report on the manuscript by Guangci Li, Xuebing Li and co-workers submitted to Catalysts.
From a general point of view, the study is well conducted. The different steps: synthesis, characterization, catalytic tests, influence of some parameters are relevant and well described.
The Introduction is a little "poor" of references, some suggestions are added below.
Otherwise, the work merit to be published in Catalysts.
In the Introduction:
when the authors cite refs 7-9 (page 1 line 37), for the interest of oligomerization of ethylene, they should add some important references belonging to this topic:
* Fliedel, C.; Ghisolfi, A.; Braunstein, P. Chem. Rev. 2016, 116 (16), 9237-9304.
* McGuinness, D. S. Chem. Rev. 2011, 111 (3), 2321-2341.
* van Leeuwen, P. W. N. M.; Clément, N. D.; Tschan, M. J. L. Coord. Chem. Rev. 2011, 255 (13-14), 1499-1517.
* Ittel, S. D.; Johnson, L. K.; Brookhart, M. Chem. Rev. 2000, 100, 1169-1203.
again line 39, homogeneous catalysts, they may site the following
* Breuil, P.-A. R.; Magna, L.; Olivier-Bourbigou, H. Catal. Lett. 2015, 145, 173-192.
* Dagorne, S.; Fliedel, C. Top. Organomet. Chem. 2013, 41, 125-171.
and refs therein.
Author Response
Dear Reviewer :
Thank you for your comments concerning our manuscript entitled “Ethylene oligomerization over nickel supported silica–alumina catalysts with high selectivity for C10+ products” (ID: catalysts-706186).
We gratefully appreciate the valuable comments and suggestions from the editor and reviewers, which significantly helped improve the quality of our work. We carefully addressed all the comments point-by-point and significantly revised our manuscript (using "Track Changes" function or marked in blue) and the SI accordingly.
Please see the attachment.
Thank you and best regards.
Yours sincerely,
Guangci Li
Corresponding author:
Email: ligc@qibebt.ac.cn
